☼ PLOS | ONE

# Experimental evidence of subtle victim blame in the absence of explicit blame

**Carolyn L. Hafer**[ID]*, **Alicia N. Rubel**[¤a], **Caroline E. Drolet**[¤b]

Department of Psychology, Brock University, St. Catharines, Ontario, Canada

¤a Current address: Faculty of Social Work, Wilfrid Laurier University, Kitchener, Ontario, Canada
¤b Current address: College of Human Medicine, Michigan State University, East Lansing, Michigan, United States of America
* chafer@brocku.ca

**Data Availability Statement:** For data and materials for all studies in this paper are publicly available and can be found at DOI 10.17605/OSF.IO/GZ3WH, https://osf.io/gz3wh/.

## Abstract

We argue that people will often eschew explicit victim blame (e.g., claiming that "X is to blame") because it is counternormative and socially undesirable, yet they might still engage in subtle victim blame by attributing victims' suffering to behaviors the victims can control (i.e., "high control causes"). We found support for this argument in three online studies with US residents. In Studies 1 and 2, participants viewed a victim posing either a high threat to the need to believe in a just world, which should heighten the motivation to engage in victim blame, or a low threat. They then rated explicit blame items and attributions for the victim's suffering. Explicit blame was low overall and not influenced by victim threat. However, participants attributed the high threat victim's suffering, more than the low threat victim's suffering, to high control causes, thus showing a subtle blame effect. In Study 2, explicit blame and subtle blame were less strongly associated (in the high threat condition) for individuals high in socially desirable responding. These results are consistent with our argument that explicit and subtle blame diverge in part due to social desirability concerns. In Study 3, most participants believed others viewed the explicit blame items, but not the attribution items, as assessing blame. Thus, attributions to high control causes can be seen as "subtle" in the sense that people believe others will view such statements as reflecting constructs other than blame. Our studies suggest a way of responding to innocent victims that could be particularly relevant in a modern context, given increasing social undesirability of various negative responses to disadvantaged and victimized individuals.

## Introduction

The phenomenon of blaming innocent victims for their fate (i.e., "victim blame") has received intensive study in several scholarly literatures (see [1–3]). Researchers propose a number of reasons for victim blame, including ideology (e.g., [4,5]) and a motivation to see the world as just or legitimate [6,7]. Across these perspectives, victim blame is typically assessed via explicit measures. In the present studies, we tested one way in which people might more subtly blame victims for their plight. Specifically, people might attribute a victim's plight to a cause that is

**Funding:** This work was supported by a grant [number 435-2014-0551] to CLH (Principal Investigator) from the Social Sciences and Humanities Research Council of Canada (http://www.sshrc-crsh.gc.ca/home-accueil-eng.aspx). The funders had no role in study design, data collection and analysis, decision to publish, or preparation of the manuscript.

**Competing interests:** The authors have declared that no competing interests exist.

perceived to be under the victim's control, even in the absence of obvious cues regarding the cause of the victim's fate.

In most victim blame studies, researchers assess explicit blame, with items such as "To what extent do you think that [victim's name]'s behavior is to blame for [victim's condition]?" (e.g., [8,9]). One problem with these measures is that recent findings from several countries [10–13] show that blaming victims for their fate is often counternormative. Thus, people might not explicitly report blame [14]. Moreover, it is likely that explicit blame has increased in social undesirability over the years, similar to certain prejudiced attitudes (see [15]). Thus, people might be increasingly reluctant to explicitly report blame. Nonetheless, perceivers' ideology or motivations could compel them to engage in victim blame. As a result, perceivers might be motivated to express victim blame in a subtler manner (see [10,12,16]). This reasoning suggests that it is important to investigate not just explicit victim blame, but also subtle forms of blame. For example, as we investigate in the current studies, people might attribute a victim's suffering to a cause that is under the victim's control; thus implying, though not explicitly stating, that the victim is to blame for his or her fate.

Only a few researchers have investigated subtle, negative appraisals of innocent victims. For example, although not assessing victim blame per se, Aguiar, Vala, Correia, and Pereira [17] and Dawtry et al. [12] examined subtle victim character derogation. Victim character derogation can be defined as assigning negative traits to a victim or reporting an overall negative opinion of a victim (as opposed to claiming the victim is to blame for his or her fate; see [7,18]). These authors found evidence of subtle character derogation under conditions that should heighten the motivation to negatively appraise the victim; specifically, when the victim posed a threat to the need to believe in a just world. The findings support justice motive theory [14,19]. According to Lerner [14], people can respond to innocent victims in a deliberative and rational manner, leading to normative reactions (e.g., sympathy). Alternatively, innocent victims can threaten a basic need to believe in a just world where people get what they deserve. In these cases, responses are more intuitive, motivated by a desire to defend a "belief in a just world," such as by negatively appraising the victim [20]. Character derogation, for example, defends a belief in a just world by portraying victims as undesirable and, thus, deserving of their suffering. Victim blame defends a belief in a just world by portraying victims as responsible for, and thus deserving of, their suffering. Given that negative appraisals of innocent victims are often counternormative, this type of intuitive and defensive response should be more likely when assessed with measures of subtle negative appraisals [12].

In the current research, we add to the nascent literature on subtle, negative appraisals of innocent victims by investigating subtle blame of innocent victims. Specifically, we examine whether, in the absence of obvious cues about cause, people attribute victims' suffering to a cause that is under the victims' control; that is, a "high control cause." For instance, a person sustains injuries from driving into a lamppost. An example of a high control cause is reckless driving, whereas an example of a less controllable cause is swerving when an animal darts into traffic. For a victim suffering concussion, playing a high-risk sport without protective headgear is a high control cause, whereas slipping on the ice outside one's home is a less controllable cause.

Research on Weiner's theory of responsibility and stigma (e.g., [21,22]) shows that attributing a victim's condition to causes that are under the victim's control predicts blaming or holding the victim responsible for his or her condition. In this research, attributions are manipulated or measured as an independent variable. In the current research, we treat attributions to high and low control causes as a dependent variable. Given that attributions to high control causes are associated with victim blame, we propose that these attributions might be used to subtly blame innocent victims for their fate. We further propose that such attributions are subtle in the sense that people do not think others will see their attributions in terms of

blame, but rather in terms of other constructs. For example, people might believe others will see these attributions as reflecting merely their knowledge of causal factors.

The closest research to the present studies was conducted by Bal and van den Bos [23]. Though their primary purpose was not to assess subtle blame, Bal and van den Bos included such a measure in their Study 1. Participants read about a car accident victim. Subtle blame was measured by asking participants how many alcoholic beverages the victim had drunk before the accident, though alcohol was not mentioned in the scenario. A greater number of drinks presumably indicated greater blame. Participants reported a greater number of drinks when threat to the need to believe in a just world, and therefore the motivation to blame or otherwise negatively appraise the victim, was heightened. (Note that this effect occurred both in a self-focus condition and a no focus condition, but not in an other-focus condition). In reporting a greater number of drinks, participants might have been attributing the victims' suffering to a high control cause—drinking alcohol—and, thus, subtly blaming the victim. Alternatively, this "subtle blame effect" could reflect a motivation to endorse any cause for the suffering of threatening victims, not especially a high control cause. Including low control causes as a control variable would help to address this alternative interpretation. Furthermore, it is unclear whether the alcohol measure was subtle in the sense that participants believed others would view it as assessing a construct other than blame (e.g., knowledge, memory), especially given that an explicit blame measure preceded the alcohol measure.

## The current studies

In the present studies, we investigated subtle victim blame from a justice motive theory perspective, as did Bal and van den Bos [23]. Our research differs from Bal and van den Bos, however, in several ways. First, we conducted a pretest to select causes of the victim's condition that are seen as high versus low in controllability. Second, we used a different manipulation of threat to the need to believe in a just world and a different victim situation. Third, we tested our argument regarding the more counternormative nature of explicit versus subtle blame by examining the role of socially desirable response tendencies in victim blame. Fourth, we examined whether the attributions to high control causes can be considered subtle blame in the sense that participants do not believe others view the attribution questions as meant to assess blame.

In Studies 1 and 2, we manipulated the threat that a victim posed to the need to believe in a just world. We then measured attributions to high control causes (i.e., subtle blame) and low control causes (a control variable), as well as explicit blame. The high threat condition should heighten the motivation to defensively blame the victim. Thus, one might expect participants to blame the victim more in the high threat than in the low threat condition. We predicted such evidence of victim blame, but only on our measure of subtle blame. Specifically, we predicted that participants in the high threat condition would attribute high control causes to the victim to a greater degree than participants in the low threat condition, thus showing a subtle blame effect; and we predicted that this effect should occur controlling for attributions to low control causes (Hypothesis 1). Note that, if the subtle blame effect reflects merely an underlying desire to find any cause for the high threat victim (rather than specifically those that place blame on the victim), then it should not occur when controlling for attributions to low control causes. Given that explicit blame is often counternormative, at least in a modern context, we predicted that participants would report low levels of explicit blame in both the high and low threat condition (Hypothesis 2).

In Study 2, we also assessed individual differences in socially desirable responding. If explicit blame is counternormative and socially undesirable, whereas attribution to high control causes (i.e., subtle blame) is not, then the relation between these two variables should be

moderated by individual differences in socially desirable responding. Our reasoning for this claim is based on the finding that social desirability concerns moderate the correlation between explicit and subtle (e.g., implicit) forms of various constructs [24]. For example, when social desirability concerns are strong, people will often eschew explicit statements of prejudiced attitudes, yet still express the attitude on measures of subtle prejudice. Thus, explicit versus subtle forms of the attitude will not be highly correlated. The correlation will be higher, however, when social desirability concerns are weak such that people are comfortable with, not only subtle, but also explicit expressions of the attitude (e.g., [25]). We apply similar reasoning to victim blame. Specifically, we argue that people will often eschew explicit blame because it is counternormative and they do not want to appear in an unfavorable light. In contrast, people will be comfortable with the subtle blame we assess in our studies, because they believe that attributions to high control causes are not viewed by others as an expression of blame. Thus, at least when social desirability concerns are strong, subtle blame and explicit blame should not be highly correlated. However, this correlation should be higher when social desirability concerns are weak. Therefore, we predicted that subtle blame (i.e., attributions to high control causes) would be associated with explicit blame to a lesser extent for participants high, compared to low, in socially desirable responding; and furthermore, that this would especially be the case in the high threat condition in which participants are particularly compelled to engage in victim blame (Hypothesis 3).

In Study 3, we tested whether attributions to high control causes can be seen as "subtle" blame in the sense that people do not think others will view such statements in terms of victim blame. Participants were asked open-ended questions about the meaning of the attribution and explicit blame items from Studies 1 and 2. We predicted that participants would view the explicit blame, but not the attribution items, as meant to assess victim blame (Hypothesis 4).

In summary, our aim was to examine one way in which people might engage in subtle victim blame; specifically, by attributing a victim's suffering to causes perceived to be under the victim's control. We conducted three studies to achieve this aim. In Study 1, we tested whether people would subtly blame the victim when their motivation to engage in victim blame is heightened, and whether this would occur in the absence of a similar effect for explicit blame. In Study 2, we attempted to replicate the findings of Study 1 and test our reasoning that explicit and subtle blame diverge due to the social undesirability of the former. In Study 3, we attempted to clarify that the form of subtle blame investigated in these studies is subtle in the sense that people do not believe others will see such attributions in terms of blame. In each study we found support for our hypotheses.

## Study 1

We report all measures, manipulations, and exclusions in these studies. In addition, sample sizes were determined before any data collection, and were based on a priori power analyses, except for Study 3, for which we recruited enough participants to yield at least 30 individuals per condition. Data and materials for all studies are publicly available and can be found at DOI 10.17605/OSF.IO/GZ3WH.

For each study in this paper, participants gave informed consent by reading an online informed consent letter, and then clicking on "Yes." All studies were reviewed and approved by Brock University's Social Science Research Ethics Board.

### Pretest

Fifty US residents were recruited over Amazon's Mechanical Turk (MTurk). After providing informed consent, participants rated the extent to which each of 12 behaviors are avoidable

and under a person's control (1 = *completely unavoidable/no control* to 7 = *completely avoidable/complete control*). Avoidability and control ratings were highly correlated (mean $r = .71$), thus we averaged them to create a composite measure of control. Participants also rated how likely they thought it was that each behavior could cause sepsis, the illness contracted by the victim in our studies (1 = *not at all likely* to 7 = *very likely*). Sepsis was defined as a bacterial infection of the blood (i.e., "blood poisoning"). Finally, participants were debriefed and paid $0.50.

We selected two high control behaviors (having unprotected sex, getting a tattoo at an unlicensed tattoo parlor) and two low control behaviors (scraping one's knee, stepping on a nail). For the high control behaviors, the average for the composite control measure ($M = 6.49$, $SD = 0.69$) was greater than the scale midpoint, $t(49) = 25.43$, $d = 3.60$, $p < .001$, indicating relatively high control. For the low control behaviors, average control ($M = 3.59$, $SD = 1.22$) was less than the midpoint, $t(49) = -2.39$, $d = -0.34$, $p = .02$, indicating relatively low control. Average control for the high control behaviors was greater than for the low control behaviors, $t(49) = 14.77$, $d = 2.09$, $p < .001$. However, the high control ($M = 4.78$, $SD = 1.31$) versus low control behaviors ($M = 5.02$, $SD = 1.31$) were seen as equally likely to cause sepsis, $t(49) = -1.28$, $d = -0.18$, $p = .21$. Finally, a principal components analysis with varimax rotation yielded two components. The low control behaviors had high loadings on Component 1 ($> .86$) and low loadings on Component 2 ($< +/-.14$), whereas the high control behaviors had high loadings on Component 2 ($> .75$) and low loadings on Component 1 ($< +/-.05$).

## Method for main study

**Participants.** We recruited 254 US residents through MTurk (136 women, 118 men; $M_{age}$ = 36.23, $SD_{age}$ = 11.87; 72% White; 48% Christian, 45% atheist/agnostic).

**Procedure and manipulation.** Participants first provided informed consent. Participants saw an initial video of a young woman who had had her legs and hands amputated due to treatment she underwent for sepsis (the cause of the sepsis was not mentioned). The woman was said to be 21 years old and about to enter nursing school. She was still in hospital and was extremely distressed. After this video, participants were told they would soon view a second video offering further information about the victim's situation. We manipulated threat to the need to believe in a just world through these instructions by varying participants' expectations about the severity of the victim's suffering (for similar manipulations, see [26],[27]). In the high just-world threat or severe suffering condition, participants were told simply that the second video gave more information about the victim's condition. These participants would presumably expect to see the victim continue to suffer. In the low just-world threat or mild suffering condition, participants were told that the second video showed the victim doing well (e.g., she had state-of-the-art prosthetics and had returned to school).

Before watching the second video, participants completed the following measures: a manipulation check; attribution items (including those assessing subtle blame), which were presented in random order; explicit victim blame items; and filler questions disguising the focus of the study (e.g., "How easy or difficult was it to understand the video?"). Attribution items preceded the explicit blame items.

All participants viewed the video described in the low threat condition. At the end of the study, participants provided demographics, and were debriefed and paid $1.00.

**Dependent variables.** As a manipulation check, participants rated how well the victim was doing (1 = *very poorly* to 7 = *very well*).

Subtle blame was assessed by having participants rate the likelihood that each of the two high control behaviors ($r = .41$) was the reason the victim had contracted sepsis (1 = *not at all*

*likely* to 7 = *very likely*). Participants similarly rated the two low control behaviors (*r* = .51). We averaged the ratings for the two high control and the two low control behaviors to yield two composite scores. The high and low control behaviors were presented along with two of the unselected behaviors from the pretest that were seen as unlikely to cause sepsis. These extra behaviors masked our focus on only a few causes.

Explicit blame was assessed with four items (e.g., "To what extent is Taylor to blame for the fact that she had her limbs amputated?"; 1 = *not at all* to 7 = *completely*). Responses were averaged to create a composite, explicit blame score (Cronbach's α = .80).

**Variables not included in the final analyses.** As part of a separate line of inquiry, we also manipulated participants' sense of purpose in life to test whether sense of purpose moderates the effect of just-world threat on reactions to victims. There were no effects involving the purpose manipulation; therefore, our analyses were conducted across this variable.

Aside from the explicit blame items, there were two items assessing character derogation and four items assessing psychological distancing from the victim and her situation. Neither of these variables was significantly affected by the suffering manipulation (or the purpose manipulation), though victim character derogation was marginally greater in the severe suffering condition (*M* = 2.54, *SD* = 1.11) than the mild suffering condition (*M* = 2.30, *SD* = 1.13), *t* (252) = -1.72, *p* = .09. Finally, responses to an open-ended question asking how the victim likely contracted sepsis were not analyzed because we failed to establish a reliable coding scheme for high versus low control causes.

## Results and discussion

**Preliminary analyses.** Participants in the mild suffering condition (*M* = 5.06, *SD* = 1.36) thought the victim was doing better than did participants in the severe suffering condition (*M* = 3.47, *SD* = 1.53), *t*(252) = 8.76, *d* = 1.10, *p* < .001. Thus, our manipulation of victim suffering (i.e., threat to the need to belief in a just world) was effective. Correlations among the variables used to test hypotheses are in Table 1.

**Victim blame.** We conducted an ANCOVA with Victim Suffering (mild vs. severe) as the independent variable and the perceived likelihood that the high control behaviors caused the victim's sepsis as the dependent variable. We controlled for the perceived likelihood that the low control behaviors caused the victim's sepsis. A sensitivity analysis showed that the ANCOVA had 80% power to detect an effect size of $\eta_p^2$ = .03, with α = .05. Consistent with Hypothesis 1, participants in the severe suffering condition rated the high control behaviors as

**Table 1. Correlations between primary variables, Study 1 (boldface, above diagonal) and Study 2 (below diagonal).**

| Variable | 1. | 2. | 3. | 4. | 5. |
|---|---|---|---|---|---|
| 1. Victim suffering | - - - - | **.03** | **.03** | **.19**\*\* | - - - - |
| 2. Explicit blame | .09 | - - - - | **.13**\* | **.30**\*\*\* | - - - - |
| 3. Low control behaviors | -.07 | .19\*\* | - - - - | **.45**\*\*\* | - - - - |
| 4. High control behaviors | .12\* | .39\*\*\* | .14\* | - - - - | - - - - |
| 5. Self-deceptive enhancement | -.07 | -.10 | -.12\* | .04 | - - - - |
| 6. Impression management | -.05 | -.10 | -.09 | .01 | .58\*\*\* |

For Study 1, *N* is 254. For Study 2, *N* is 295 or 296. Low control behaviors and high control behaviors = perceived likelihood that low or high control behaviors led to the victim's illness. Victim suffering = severe suffering (1) vs. mild suffering (0).

\**p* < .05.

\*\**p* < .01.

\*\*\**p* < .001.

more likely to have caused the victim's sepsis ($M$ = 3.44, $SD$ = 1.35) than did participants in the mild suffering condition ($M$ = 2.95, $SD$ = 1.21), $F$(1, 251) = 9.55, $p$ = .002, $\eta_p^2$ = .04, 95% $CI_{mean\ diff}$ [0.16, 0.73]. An ancillary ANCOVA showed that ratings for the low control behaviors (controlling for high control behaviors) did not differ by condition (overall $M$ = 3.83, $SD$ = 1.36), $F$(1, 251) = 0.83, $p$ = .37, $\eta_p^2$ = .003, 95% $CI_{mean\ diff}$ [-0.45, 0.17]. An ancillary Victim Suffering (mild vs. severe) X Controllability of Causes (low vs. high control behaviors) mixed ANOVA yielded a significant interaction, $F$(1, 252) = 5.07, $p$ = .03, $\eta_p^2$ = .02, showing that victim suffering affected the ratings for high control behaviors, $p$ = .003, $d$ = 0.38, 95% $CI_{mean\ diff}$ [0.17, 0.80], independent of the (null) effect on ratings for low control behaviors, $p$ = .61, $d$ = 0.07, 95% $CI_{mean\ diff}$ [-0.25, 0.43]. Details for the mixed ANOVA are in the Supporting Information (S1 Text). Taken together, these findings suggest that participants were not more motivated to find any kind of cause for the high threat versus low threat victim's suffering. Rather, they were motivated to more strongly endorse a high control cause, despite no obvious cues in the video as to the cause of the illness. We believe these findings reflect a motivation to subtly blame a threatening victim by attributing the victim's suffering to a cause that is under the victim's control.

We used a $t$ test to test Hypothesis 2. A sensitivity analysis showed that the $t$ test had 80% power to detect an effect size of $d$ = 0.35, with $\alpha$ = .05. Victim suffering did not affect explicit blame, $t$(252) = .44, $d$ = 0.06, $p$ = .66, 95% CI [-0.20, 0.31], and explicit blame was low overall (overall $M$ = 2.33, $SD$ = 1.03). Thus, as hypothesized, participants did not appear to resolve the threat posed by the severely suffering victim by explicitly blaming her, presumably because of the counternormative nature of victim blame.

In summary, we found evidence that, under conditions that should heighten the desire to engage in victim blame, people will subtly blame a victim by attributing the victim's suffering to a behavior that the victim could have avoided. Furthermore, our findings suggest that this subtle blame can occur in the absence of similarly motivated, explicit blame.

## Study 2

For Study 2, we attempted to replicate the effect of victim suffering on the perceived likelihood that high control behaviors caused the victim's illness. We also tested Hypothesis 3: Subtle blame (i.e., attributions to high control causes) will be less strongly associated with explicit blame for participants high versus low in socially desirable responding, especially in the severe suffering (i.e., high threat) condition.

### Method

**Participants.** We recruited 300 US residents through MTurk. Four participants' data were removed—two in the severe and one in the mild suffering condition—because they did not watch the complete videos, leaving 296 cases (141 women, 154 men, 1 unspecified; $M_{age}$ = 36.54, $SD_{age}$ = 12.87; 78% White; 48% Christian, 44% atheist/agnostic).

**Procedure.** The procedure, manipulation, and measures were similar to those for Study 1. However, we omitted the Study 1 "variables not included in the analysis." We added Paulhus's [28] 40-item Balanced Inventory of Desirable Responding (BIDR), which participants completed immediately after giving informed consent and before watching the first video. The BIDR contains two subscales: impression management, or deliberately responding in way to make a favorable impression on others (e.g., "I never swear"); and self-deceptive enhancement, or possessing an overly positive view of the self (e.g., "I am a completely rational person"). Items were rated from 1 = *not true* to 7 = *very true*. After reverse-scoring appropriate items, participants received 1 point for each "6" or "7." We summed these points separately for the

impression management ($\alpha$ = .85) and self-deceptive enhancement ($\alpha$ = .82) subscales (for more information on this scoring, see [28]).

## Results

**Preliminary analyses.** Participants in the mild suffering condition ($M$ = 5.35, $SD$ = 1.29) thought the victim was doing better compared to participants in the severe suffering condition ($M$ = 3.40, $SD$ = 1.51), $t(294)$ = 12.02, $d$ = 1.39, $p$ < .001. Thus, the manipulation of victim suffering (i.e., threat to the need to believe in a just world) was effective. Correlations among the variables used to test hypotheses are in Table 1.

**Victim blame.** An ANCOVA with Victim Suffering (mild vs. severe) as the independent variable was conducted on the perceived likelihood that the high control behaviors caused the victim to contract sepsis. We controlled for the perceived likelihood that the low control behaviors caused the victim's sepsis. The analysis had 80% power to detect an effect size of $\eta_p^2$ = .03, with $\alpha$ = .05. Consistent with Hypothesis 1, participants in the severe suffering condition rated the high control behaviors as more likely to have caused the victim's sepsis ($M$ = 3.12, $SD$ = 1.27) than did participants in the mild suffering condition ($M$ = 2.82, $SD$ = 1.18), $F(1, 292)$ = 5.65, $p$ = .02, $\eta_p^2$ = .02, 95% CI$_{mean\ diff}$ [0.06, 0.61]. An ancillary ANCOVA showed that ratings for the low control behaviors (controlling for high control behaviors) did not differ by condition, (overall $M$ = 3.71, $SD$ = 1.36), $F(1, 292)$ = 2.55, $p$ = .11, $\eta_p^2$ = .01, 95% CI$_{mean\ diff}$ [-0.56, 0.06]. An ancillary mixed ANOVA yielded a significant Victim Suffering X Controllability of Causes interaction, $F(1, 293)$ = 6.73, $p$ = .01, $\eta_p^2$ = .02, showing that ratings for high control behaviors were affected by victim suffering, $p$ = .03, $d$ = 0.24, 95% CI [0.03, 0.59], independent of the (null) effect on low control behaviors, $p$ = .21, $d$ = -0.15, 95% CI [-0.51, 0.11]. Details are in the Supporting Information (S2 Text).

A $t$ test used to test Hypothesis 2 had 80% power to detect an effect size of $d$ = 0.33, with $\alpha$ = .05. As predicted, victim suffering did not affect explicit blame (overall $M$ = 2.32, $SD$ = 0.99), $t(294)$ = 1.53, $d$ = 0.18, $p$ = .13, 95% CI [-0.05, 0.40].

**Social desirability.** To test Hypothesis 3, we ran one hierarchical regression for each BIDR subscale. The victim suffering manipulation, individual differences in socially desirable responding (centered), subtle blame (the perceived likelihood that high control behaviors caused the victim's situation; centered), and the perceived likelihood ratings of low control behaviors were entered on the first step; the two-way interactions were entered on the second step; and the three-way interaction was entered on the third step. Explicit blame was the criterion. The analyses had 80% power to detect an effect size of $f^2$ = .03, with $\alpha$ = .05, and 8 predictors.

We report only interactions from the regressions. The analysis involving the self-deceptive enhancement subscale yielded a marginal Self-Deceptive Enhancement X Subtle Blame interaction, $b$ = -0.34, $t(287)$ = -1.77, $p$ = .08, $sr^2$ = .01, 95% CI [-0.73, 0.04]. Though only marginal, the pattern is consistent with Hypothesis 3, in that subtle blame predicted explicit blame to a lesser extent for individuals high ($M$ + 1 $SD$), $b$ = 0.25, $t(287)$ = 3.55, $p$ < .001, $sr^2$ = .03, 95% CI [0.11, 0.38] versus low ($M$—1 $SD$), $b$ = 0.39, $t(287)$ = 4.72, $p$ < .001, $sr^2$ = .06, 95% CI [0.23, 0.55], in self-deceptive enhancement. Moreover, this interaction was superseded by a significant three-way interaction, $b$ = -0.77, $t(286)$ = -2.00, $p$ = .046, $sr^2$ = .01, 95% CI [-1.53, -0.01] that was also consistent with Hypothesis 3 (see Figs 1 and 2). In decomposing this interaction, we found that the Self-Deceptive Enhancement X Subtle Blame interaction was significant in the severe suffering, $b$ = -0.75, $t(286)$ = -2.68, $p$ = .008, 95% CI [-1.29, -0.20], but not the mild suffering, $b$ = 0.02, $t(286)$ = .09, $p$ = .93, 95% CI [-0.50, 0.55], condition. Within the severe condition, again subtle blame predicted explicit blame to a lesser extent for individuals high,

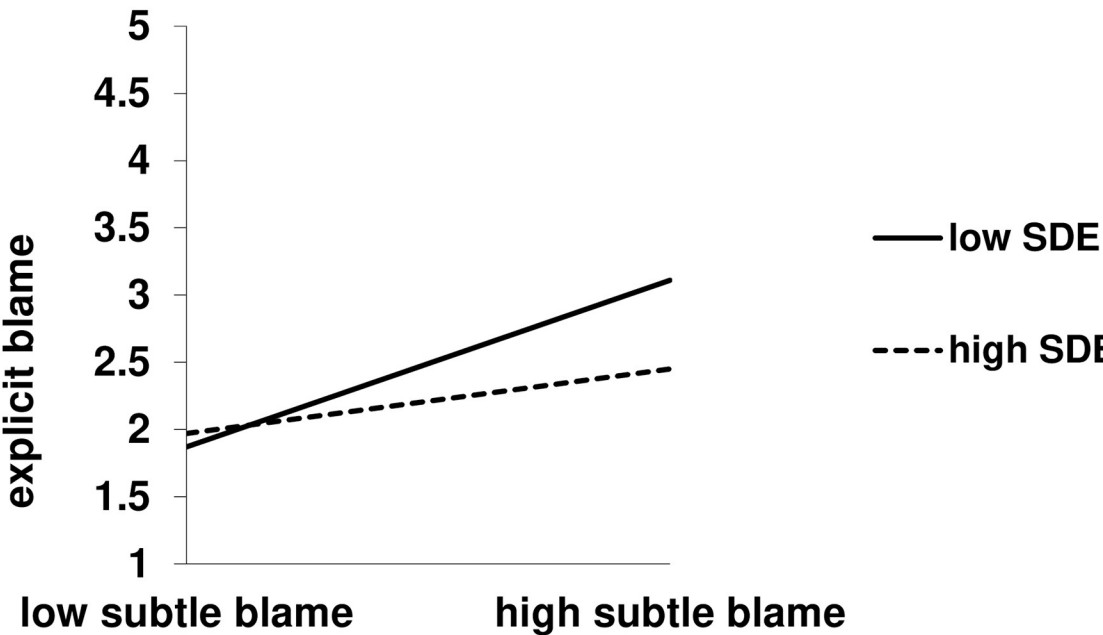

**Fig 1. Self-Deceptive Enhancement (SDE) X subtle blame interaction on explicit blame, severe suffering condition.** Subtle blame = perceived likelihood of high control causes. Results are controlling for perceived likelihood of low control causes.

$b = 0.20$, $t(286) = 2.71$, $p = .007$, $sr^2 = .02$, 95% CI [0.05, 0.34], versus low, $b = 0.50$, $t(286) = 5.50$, $p < .001$, $sr^2 = .08$, 95% CI [0.32, 0.68], in self-deceptive enhancement. The analysis involving the impression management subscale yielded no interactions, all $ps > .23$. Detailed statistics for the regressions are in the Supporting Information (S1 and S2 Tables).

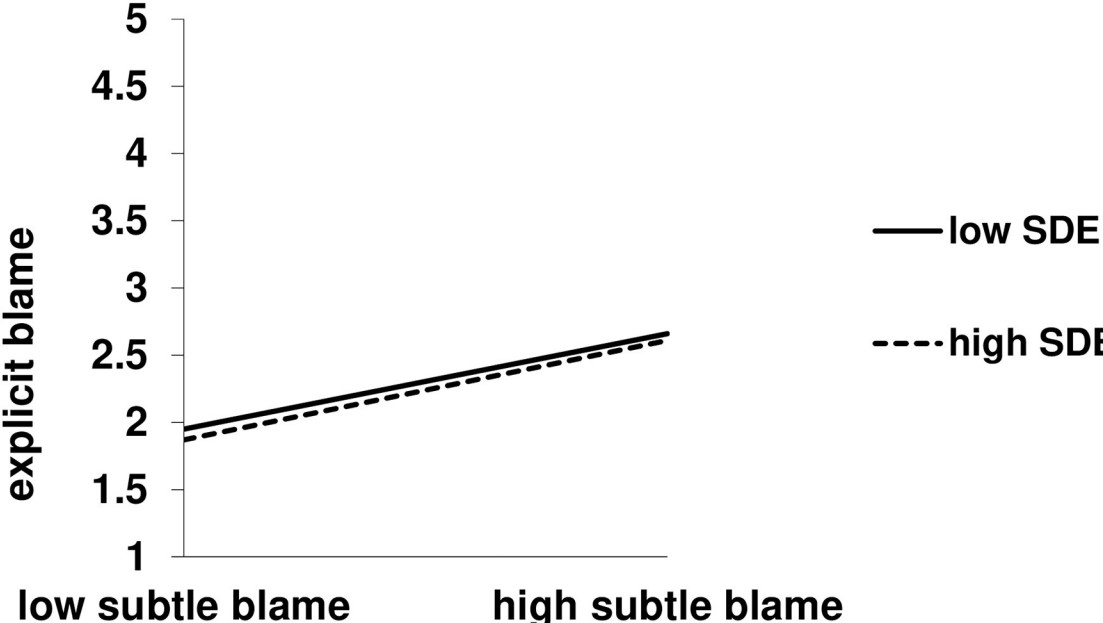

**Fig 2. Self-Deceptive Enhancement (SDE) X subtle blame interaction on explicit blame, mild suffering condition.** Subtle blame = perceived likelihood of high control causes. Results are controlling for the perceived likelihood of low control causes.

## Discussion

We again found evidence that, when the motivation to engage in victim blame is heightened, people will subtly blame victims by attributing the victims' suffering to a cause that is perceived to be under their control.

We also replicated evidence for a subtle blame effect in the absence of a similar effect on explicit blame. Our interpretation is that perceivers will often resort to subtle blame when compelled to engage in victim blame because explicit blame is counternormative and thus inhibited; whereas subtle blame, not being viewed as an expression of victim blame, is not inhibited. Our individual difference analyses are consistent with this argument, as well as with research on the relation between subtle (e.g., implicit) and explicit constructs (see [24]). Specifically, in the high threat condition, explicit blame was less strongly associated with subtle blame for participants high in self-deceptive enhancement, a form of socially desirable responding.

## Study 3

Studies 1 and 2 show evidence of subtle victim blame via the attribution of high control causes. We argued that these attributions constitute subtle blame in the sense that participants do not think the attribution items are meant to assess blame, and thus are not viewed as victim blame in the eyes of others. In Study 3, we asked participants open-ended questions about the meaning of either the explicit blame or the attribution items. If our claim is correct, consistent with Hypothesis 4, participants given the explicit items should be more likely to say the items assess blame, whereas participants given the attribution items should be more likely to say the items assess other constructs.

## Participants

We recruited 64 US residents through MTurk (38 women, 26 men; $M_{age}$ = 37.22, $SD_{age}$ = 11.47; 77% White; 49% Christian, 41% atheist/agnostic).

## Procedure

After providing consent, participants viewed the severe suffering (i.e., high just-world threat) video. They then answered filler items followed by either the explicit blame or the high and low control attribution items from Studies 1 and 2. Following the blame/attribution items, participants answered two open-ended questions: "Why do you think we are asking this question?," and "What do you think this question is measuring?" Participants then viewed the video in which the victim was doing well, provided demographics, and were debriefed and paid $1.00.

Two individuals who were blind to participants' condition independently coded the open-ended responses into those reflecting "blame and related constructs," and "other." Interrater reliabilities were = .96 for the first open-ended question, and = .87 for the second question. Disagreements were resolved through discussion. The most common "other" responses indicated that the study or measures assessed health knowledge (32% of "other" responses for the first open-ended question, 45% for the second question) or indicated uncertainty (29% of "other" responses for the first question, 20% for the second question).

## Results and discussion

We performed a 2 (measure: explicit blame, attributions) X 2 (category: blame, other) contingency table analysis for each open-ended question. The analysis had 80% power to detect an

**Table 2. Open-ended responses, Study 3.**

| Type of Measure | "Why do you think we are asking this question?" | | "What do you think this question is measuring?" | |
|---|---|---|---|---|
| | Blame Response | Other Response | Blame Response | Other Response |
| Explicit Blame Scale | | | | |
| *n* | 19 | 12 | 20 | 12 |
| % within measure | 61.3 | 38.7 | 62.5 | 37.5 |
| % within response category | 86.4 | 29.3 | 83.3 | 30.0 |
| Standardized residual | 2.5 | -1.8 | 2.3 | -1.8 |
| Attributions | | | | |
| *n* | 3 | 29 | 4 | 28 |
| % within measure | 9.4 | 90.6 | 12.5 | 87.5 |
| % within response category | 13.6 | 70.7 | 16.7 | 70.0 |
| Standardized residual | -2.4 | 1.8 | -2.3 | 1.8 |

effect size of $w = .35$, with $\alpha = .05$, $df = 1$. For "Why do you think we are asking this question?," $\chi^2(1, N = 63) = 18.67$, $p < .001$, phi = -.54, and "What do you think this question is measuring?," $\chi^2(1, N = 64) = 17.07$, $p < .001$, phi = -.52, the proportion of participants endorsing each response category varied by the type of measure (see Table 2). As hypothesized, most participants who were given the explicit blame measure mentioned blame or some related concept in their open-ended responses, whereas most participants who received the attribution items mentioned a construct other than blame. Thus, unlike explicit statements of blame, people do not appear to believe that others (e.g., the researchers) view such behavioral attributions for a victim's suffering in terms of blame. Attributions to high control causes can therefore be labelled "subtle" blame in this sense.

## General discussion

Victim blame has a long history of study in the social sciences, with the vast majority of studies focussed on explicit blame. We investigated one way in which people more subtly engage in victim blame—by attributing victims' fate to a cause that is under their control. In two experiments (Studies 1 & 2), we found evidence of this phenomenon. Supporting Hypothesis 1, participants exposed to a victim who presumably threatened a need to see the world as just, compared to participants exposed to a less threatening victim, reported a greater likelihood that the victim had contracted her illness through behaviors over which she had a high degree of control. Supporting Hypothesis 2, this effect occurred despite no similar effects for explicit blame (which was low overall), and despite no obvious cues regarding the cause of the victim's illness. Our interpretation of these findings is that explicitly blaming innocent victims for their fate is often counternormative (e.g., [10–12]), inhibiting explicit blame. Thus, when compelled to blame victims—in the present case, when a motivation to defensively blame victims is heightened by a threat to the need to believe in a just world—people might resort to attributions to high control causes as a subtle form of blame.

These studies add to previous work by Bal and van den Bos [23]. Bal and van den Bos measured attributions to a high control cause; however, we also measured attributions to low control causes as a covariate. The effect of victim threat on attributions to high control causes occurred controlling for attributions to low control causes, and victim threat did not influence the latter. Thus, participants were not simply more motivated to find any cause in the high threat condition, which was an alternative explanation for the Bal and van den Bos study.

Also unlike Bal and van den Bos [23], we measured individual differences in socially desirable responding. If explicit blame is counternormative and socially undesirable, whereas attribution to high control causes (i.e., subtle blame) is not, then consistent with research on the relation between subtle and explicit constructs, the association between these two variables should be moderated by individual differences in socially desirable responding. Indeed, in Study 2 we found that, when compelled to engage in victim blame, explicit and subtle blame were less strongly associated for people who are more concerned with social desirability. These findings support Hypothesis 3 and bolster our claim that explicit and subtle blame diverge due to the counternormative nature of the former.

We argue that the blame expressed in attributions to high control causes is subtle because perceivers' causal statements about an individual's suffering can be construed as reflecting a construct other than blame; in the present case, perceivers' objective knowledge of the victim's affliction. Indeed, supporting Hypothesis 4, results from Study 3 suggested that participants believed others view the attributions as indicative of constructs other than their tendency to blame the victim; most notably, their health knowledge. Although attributions can partly reflect knowledge, perceivers can also hide a desire to blame victims behind the mantle of expressing their knowledge of the causes of various conditions (see [12]).

## Implications

Our findings extend justice motive theory by suggesting a strategy for defending a belief in a just world other than those commonly investigated (see [18]). As such, our studies add to other recent attempts to examine alternative ways people cope with threats to the need to believe in a just world (see [20]). Moreover, subtle blame in the absence of explicit blame might be a more likely response in contemporary research, if explicit blame has increased in social undesirability over the years (note that studies demonstrating that victim blame is counternormative are all recent; i.e., [10–13]), similar to certain prejudiced attitudes (see [15]).

The implications of our research reach beyond justice motive theory. We examined victim blame in the context of a threat to the need to believe in a just world. However, we expect that subtle blame, in the absence of explicit blame, could also occur under other forces that compel individuals to engage in victim blame, as long as explicit blame is viewed as counternormative. Other forces include a heightened desire to engage in system justification [6], certain ideologies (e.g., [4]), and so on.

Our findings also add to the general literature on subtle versus explicit judgments. For example, in the attitude literature, there is evidence of greater correspondence between implicit evaluations (a form of subtle judgment) and explicit evaluations when concerns about self-presentation are weak (e.g., [24]). In the literature on reactions to victims, Dawtry et al. [12] found that motivation to avoid negative responses to innocent victims moderated the degree to which participants engaged in more versus less explicit character derogation. Our findings involving social desirability are similar to those in these previous studies, but involve novel dependent variables—subtle and explicit victim blame. Furthermore, our results add specifically to Dawtry et al. [12] by suggesting that general individual differences in socially desirable responding (rather than only issue-specific differences) can moderate the relation between explicit versus subtle appraisals of victims (c.f., [25]).

From a practical perspective, our results suggest that, in the absence of explicit blame, innocent victims might yet be subject to subtle blame. Interestingly, subtle blame could be as damaging to victims as explicit blame, given that attributions to high control causes can negatively affect the desire to help victims [22]. Thus, researchers should examine ways to reduce subtle blame of innocent victims.

## Limitations and future directions

The extremity of our victim stimulus (a woman who suffered amputations) likely maximized the chance of certain results. The more extreme the victimization, perhaps the more counter-normative is victim blame, and thus the more people will express subtle blame in the absence of explicit blame. Aguiar et al. [17] found no evidence of explicit blame or character derogation for a similarly extreme victim scenario (a child who lost his arms); although, in a second study they found evidence of subtle character derogation. Note that our findings should not extend to victims for whom blame is normative.

It is unclear whether our findings generalize to other victim situations. However, we believe that our findings would generalize beyond our specific scenario given that Bal and van den Bos [23] found similar effects of a just-world threat manipulation on measures of blame, but with a different victim and a different manipulation of threat to the need to believe in a just world.

Our findings do not address whether endorsing high control causes as a form of subtle blame is a conscious or unconscious act of victim blaming. In Study 2, participants prone to self-deception (i.e., high in the self-deceptive enhancement form of socially desirable respond-ing) showed greater divergence of explicit and subtle blame. Thus, perhaps people engaging in subtle blame via attributions to high control causes believe others will not see their attributions as blaming the victim (as suggested by our Study 3), and do not consciously view their apprais-als as victim blame. Furthermore, the fact that we found moderation by self-deceptive enhancement but not impression management presents the intriguing possibility that people avoid explicit blame primarily due to a tendency to see themselves in an overly positive light rather than a desire to make a good impression on others. These issues warrant further investigation.

## Conclusion

In conclusion, victim blame can come in many forms. We found evidence that people will eschew explicitly blaming a victim yet, when compelled by motivational forces, will still subtly blame that individual for his or her fate. Interestingly, if explicit victim blame is becoming more counternormative, the prevalence of subtle blame in the absence of explicit blame could increase.

## Supporting information

**S1 Table. Hierarchical regression with self-deceptive enhancement as the measure of socially desirable responding, Study 2.**
(DOCX)

**S2 Table. Hierarchical regression with impression management as the measure of socially desirable responding, Study 2.**
(DOCX)

**S1 Text. Mixed ANOVA, Study 1.**
(DOCX)

**S2 Text. Mixed ANOVA, Study 2.**
(DOCX)

## Acknowledgments

We thank Tyler Burleigh for his help with programming.

## Author Contributions

**Conceptualization:** Carolyn L. Hafer, Alicia N. Rubel, Caroline E. Drolet.

**Data curation:** Alicia N. Rubel, Caroline E. Drolet.

**Formal analysis:** Carolyn L. Hafer, Alicia N. Rubel.

**Funding acquisition:** Carolyn L. Hafer.

**Investigation:** Alicia N. Rubel.

**Methodology:** Carolyn L. Hafer, Alicia N. Rubel, Caroline E. Drolet.

**Project administration:** Carolyn L. Hafer.

**Supervision:** Carolyn L. Hafer.

**Writing – original draft:** Carolyn L. Hafer.

**Writing – review & editing:** Carolyn L. Hafer, Alicia N. Rubel, Caroline E. Drolet.

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
