## [Decision Letter · Decision Letter 0]

31 Oct 2019

PONE-D-19-24829

Experimental evidence of subtle victim blame in the absence of explicit blame

PLOS ONE

Dear Dr. Hafer,

Thank you for submitting your manuscript to PLOS ONE. After careful consideration, we feel that it has merit but does not fully meet PLOS ONE’s publication criteria as it currently stands. Therefore, we invite you to submit a revised version of the manuscript that addresses the points raised during the review process.

I have received reviews from two experts related to justice reasoning and responses to victims. As you can see from their comments below, both reviewers had very positive things to say about your manuscript. I also read the paper prior to reading their reviews to form my own independent impression and my response mirrored that of the reviewers. Most of the suggestions the reviewers made were fairly minor, so I will not restate them here. Reviewer 1, for example, thought more discussion of why the “self-deception” subscale, specifically, of the BIDR moderated the effects would be beneficial. Similarly, Reviewer 2 wondered if some examples of high versus low control causes might be helpful in the introduction.

We would appreciate receiving your revised manuscript by Dec 15 2019 11:59PM. To enhance the reproducibility of your results, we recommend that if applicable you deposit your laboratory protocols in protocols.io, where a protocol can be assigned its own identifier (DOI) such that it can be cited independently in the future. For instructions see: http://journals.plos.org/plosone/s/submission-guidelines#loc-laboratory-protocols

We look forward to receiving your revised manuscript.

Kind regards,

Daniel Wisneski

Academic Editor

PLOS ONE

Journal Requirements:

Reviewers' comments:

Reviewer's Responses to Questions

**Comments to the Author**

1. Is the manuscript technically sound, and do the data support the conclusions?

Reviewer #1: Yes

Reviewer #2: Yes

2. Has the statistical analysis been performed appropriately and rigorously? 

Reviewer #1: Yes

Reviewer #2: Yes

3. Have the authors made all data underlying the findings in their manuscript fully available?

Reviewer #1: Yes

Reviewer #2: Yes

4. Is the manuscript presented in an intelligible fashion and written in standard English?

Reviewer #1: Yes

Reviewer #2: Yes

5. Review Comments to the Author

Reviewer #1: Thank you for the opportunity to review this paper. Across three studies, the authors investigated whether people engage in subtle victim blaming by attributing victims’ suffering to high control behaviors despite eschewing overt blame, and whether this is explained by social desirability concerns (because overtly blaming victims is counter-normative).

The findings complement and extend recent research on overt and covert responses to victims and other stigmatized groups. This is an interesting topic, and the findings have important methodological (in terms of how responses to victims are measured/what we can expect given different means of measuring them) and theoretical implications.

Overall, the paper is well written and a pleasure to read. The studies are well and designed, complementary to each other, and the findings clearly support for the authors’ arguments and hypotheses. I have only a few, relatively minor suggestions.

- Effect of suffering on attributions to low and high control causes in Studies 1 and 2. I appreciate the appropriateness of ANCOVA given hypothesis 1. It would be informative to show also whether the effect of suffering on low and high control causes differs in the way that the ANCOVA’s (on high control) and t-test’s (low control) might imply, but do not confirm – i.e., whether there is an interaction such that suffering increases attributions to high (compared to low) control causes more strongly. The results clearly suggest that greater suffering motivated ppts. to find fault with the victim’s behavior in particular, rather than more strongly endorse any old cause. If suffering does increase high control attributions significantly over and above low control attributions, I think a stronger conclusion can be drawn. These additional analyses could be reported briefly in a footnote or supplementary materials.

- Self-deception vs. impression management. In Study 2, the relation between implicit and explicit blame was moderated (under high suffering) by the self-deception, but not the impression management subscale, of the BIDR. The pattern of results supports the idea that people resort to subtle over overt blame due to social desirability concerns in general. Yet, might this pattern also suggest something about the specific locus of these concerns? My understanding is that the self-deception subscale measures the tendency to maintain a positive self-image (I don’t want to see myself as counter-normative/a person who blames victims), whereas the impression management subscale pertains to managing others’ impressions of the self (I don’t want to be seen by others as behaving counter-normatively). Space permitting, I think it would be good to briefly address this.

Minor Comments

- It would be helpful to provide intercorrelations between measures in Study 1

- There is a (presumed) typo on line 177 (data collection, rather than analysis?)

- A reference is needed on line 384 (research on relation between explicit/implicit constructs)

Reviewer #2: Review of PONE-19-24829

The authors report three studies testing subtle victim blame effects. In Study 1, they found that under conditions of just world threat, participants subtlety blamed the victim by attributing the victim’s suffering to a behaviour that the victim could have avoided (high control cause). Study 2 found that the Study 1 effect was modulated by self-deception. Study 3 shed light on the idea that people don’t perceive subtle blame as blame (vs. explicit blame judgments).

Taken together, the studies are providing important new insights into how people respond to the suffering of innocent victims. Although some recent work has begun to explore similar ideas in the context of subtle victim derogation (e.g., Dawtry et al.), none have examined how subtle victim blame can manifest as one response to the suffering of an innocent victims. The authors show that one way such subtle blame can manifest is through people endorsing causes for someone’s suffering that don’t, prima facie, seem like blame judgments.

I only have a few suggestions that might help improve the work:

1) I think some concrete examples of high vs. low control causes might help provide the reader with a bit more context for the work. Somewhere around line 83.

2) Lerner’s theorizing about the two forms of the justice motive—basically, implicit vs. explicit--doesn’t seem to be given much coverage here. It might not be exactly relevant to the current work, but mention of this theorizing might provide a bit more theoretical grounding for the research.

3) I wondered whether it would have been more appropriate to control for high control variables when testing the effect of suffering status on low control behaviors (as was done for low control behaviors)

4) Most of the theorizing in the introduction about the role of SDR was in general terms but the authors analysed impression management and self-deception separately. I think this is fine given the known two subcomponents of the BIDR, but it would be good to see more discussion of why the effects were observed for SDE and not impression management. What are the differences between these forms of SDR that might shed light on why effects were observed for one and not the other?

Signed review

Mitch Callan

6. PLOS authors have the option to publish the peer review history of their article (what does this mean?). If published, this will include your full peer review and any attached files.

Reviewer #1: No

Reviewer #2: No

---

## [Author Response · Author response to Decision Letter 0]

6 Dec 2019

Reviewer #1

1) - Effect of suffering on attributions to low and high control causes in Studies 1 and 2. I appreciate the appropriateness of ANCOVA given hypothesis 1. It would be informative to show also whether the effect of suffering on low and high control causes differs in the way that the ANCOVA’s (on high control) and t-test’s (low control) might imply, but do not confirm – i.e., whether there is an interaction such that suffering increases attributions to high (compared to low) control causes more strongly. The results clearly suggest that greater suffering motivated ppts. to find fault with the victim’s behavior in particular, rather than more strongly endorse any old cause. If suffering does increase high control attributions significantly over and above low control attributions, I think a stronger conclusion can be drawn. These additional analyses could be reported briefly in a footnote or supplementary materials.

Our response: The analysis suggested by Reviewer #1 is a Victim Suffering (mild vs. severe) X Controllability of Causes (low control vs. high control behaviors) mixed ANOVA with repeated measures on the second factor. The dependent variable is the perceived likelihood that the behaviors caused the victim’s sepsis. We conducted this mixed ANOVA for Studies 1 and 2. In both cases, there is a significant interaction between victim suffering and controllability of causes: Participants in the severe suffering condition rated the high control behaviors as more likely to have caused the victim’s sepsis than did participants in the mild suffering condition, whereas ratings for the low control behaviors did not differ by condition. Thus, as Reviewer #1 suggests should be the case, greater suffering increases high control attributions significantly, independent of any such effect on low control attributions. In the revised manuscript, we briefly state the findings of these mixed ANOVAs in the first paragraph of the “Victim blame” section of the Results for Studies 1 and 2. We refer to new supplemental information files (S3_Mixed ANOVA, S4_Mixed ANOVA) for details.

2) - Self-deception vs. impression management. In Study 2, the relation between implicit and explicit blame was moderated (under high suffering) by the self-deception, but not the impression management subscale, of the BIDR. The pattern of results supports the idea that people resort to subtle over overt blame due to social desirability concerns in general. Yet, might this pattern also suggest something about the specific locus of these concerns? My understanding is that the self-deception subscale measures the tendency to maintain a positive self-image (I don’t want to see myself as counter-normative/a person who blames victims), whereas the impression management subscale pertains to managing others’ impressions of the self (I don’t want to be seen by others as behaving counter-normatively). Space permitting, I think it would be good to briefly address this.

Our response: Reviewer 1’s point is related to our comment at the end of the “Limitations and future directions” section regarding the conscious versus unconscious nature of subtle blame. Thus, we added a sentence to this paragraph stating one implication of our finding that only self-deceptive enhancement moderated the relation between subtle and explicit blame.

3) - It would be helpful to provide intercorrelations between measures in Study 1

Our response: We revised Table 1 so that it now contains intercorrelations for Study 1 and Study 2. We also moved Table 1 to immediately after its first mention (the end of the 1st paragraph for the Results and Discussion of Study 1). 

4) - There is a (presumed) typo on line 177 (data collection, rather than analysis?)

Our response: The Reviewer is correct. We have changed “data analysis” to “data collection” (see the 1st paragraph of Study 1).

5) - A reference is needed on line 384 (research on relation between explicit/implicit constructs)

Our response: We have added a citation (see 2nd paragraph of the Discussion for Study 2).

Reviewer #2

1) I think some concrete examples of high vs. low control causes might help provide the reader with a bit more context for the work. Somewhere around line 83.

Our response: We have included two examples of high and low control causes (see the end of the 4th paragraph of the Introduction).

2) Lerner’s theorizing about the two forms of the justice motive—basically, implicit vs. explicit--doesn’t seem to be given much coverage here. It might not be exactly relevant to the current work, but mention of this theorizing might provide a bit more theoretical grounding for the research.

Our response: We have added Lerner’s notion of deliberative/rational versus intuitive/defensive processes with respect to the justice motive (see the 3rd paragraph of the Introduction).

3) I wondered whether it would have been more appropriate to control for high control variables when testing the effect of suffering status on low control behaviors (as was done for low control behaviors)

Our response: We have replaced the t tests using low control behaviors as the dependent variable with ANCOVAs, as Reviewer #2 suggests (see the 1st paragraph of the “Victim blame” section of the Results for Studies 1 and 2).

4) Most of the theorizing in the introduction about the role of SDR was in general terms but the authors analysed impression management and self-deception separately. I think this is fine given the known two subcomponents of the BIDR, but it would be good to see more discussion of why the effects were observed for SDE and not impression management. What are the differences between these forms of SDR that might shed light on why effects were observed for one and not the other?

Our response: See our response to Reviewer #1’s Point 2.

---

## [Editor Report · Decision Letter 1]

16 Dec 2019

Experimental evidence of subtle victim blame in the absence of explicit blame

PONE-D-19-24829R1

Dear Dr. Hafer,

We are pleased to inform you that your manuscript has been judged scientifically suitable for publication and will be formally accepted for publication once it complies with all outstanding technical requirements.

With kind regards,

Daniel Wisneski

Academic Editor

PLOS ONE
---

## [Editor Report · Acceptance letter]

18 Dec 2019

PONE-D-19-24829R1 

Experimental evidence of subtle victim blame in the absence of explicit blame 

Dear Dr. Hafer:

I am pleased to inform you that your manuscript has been deemed suitable for publication in PLOS ONE. Congratulations! Your manuscript is now with our production department. 

With kind regards,

on behalf of

Dr. Daniel Wisneski 

Academic Editor

PLOS ONE